# To Change or Not to Change: A Study of Workplace Change during the COVID-19 Pandemic

**DOI:** 10.3390/ijerph19041982

**Published:** 2022-02-10

**Authors:** Shu Da, Silje Fossum Fladmark, Irina Wara, Marit Christensen, Siw Tone Innstrand

**Affiliations:** 1Beijing Key Laboratory of Applied Experimental Psychology, National Demonstration Center for Experimental Psychology Education (Beijing Normal University), Faculty of Psychology, Beijing Normal University, Beijing 100875, China; zhuriyinv@163.com; 2Department of Psychology, Norwegian University of Science and Technology, N-7941 Trondheim, Norway; irinasw@stud.ntnu.no (I.W.); marit.christensen@ntnu.no (M.C.); siw.tone.innstrand@ntnu.no (S.T.I.)

**Keywords:** COVID-19 pandemic, work from home, job demands, job resources, work engagement, burnout

## Abstract

After the outbreak of the COVID-19 pandemic, many employees were suddenly required to work more from home. Previous literature on working from home may not be applicable to this mandatory and overall change. In this study, we drew on the Job Demands–Resources (JD-R) model to explore the relationships between job demands (workload and work–home conflict) as well as resources (support from leaders, coworkers, and the family) and wellbeing (burnout and work engagement) in employees who still went to the workplace (no-change group) and employees who transitioned into working from home (change group) during the COVID-19 pandemic. Data were analyzed with multivariate structural equation modeling. The results indicate that work–home conflict was detrimental for employee wellbeing in both groups. Interestingly, the workload seems to contribute to work engagement for employees who worked from home. Regarding the resources, the three different sources of social support, leaders, coworkers, and family, were all related to employee wellbeing, but in different ways. It seemed that family support was most important for employees’ wellbeing in the change group. This study presents implications for the wellbeing of employees in both the change and no-change group during the COVID-19 pandemic, emphasizing the importance of family-friendly policies.

## 1. Introduction

Working From Home (WFH), which refers to work performed at a remote location (such as home) [1], is not a novel concept. The development of information technology and a multitude of internet-based platforms has made team collaboration and distant communication between leaders and employees and between employees and their clients increasingly convenient. The benefits of WFH may include increased job satisfaction, reduced travel time and expenses, increased productivity, and reduced turnover and absenteeism [2,3,4,5]. The drawbacks of WFH may include isolation from the work culture, potential conflicts between work and home, a lack of control over employees, difficulties in teamwork, and so on [2]. A meta-analysis [3] demonstrated that WFH had small but mainly beneficial effects on proximal outcomes, such as perceived autonomy and (decreased) work–home conflict. WFH also had beneficial effects on more distal outcomes, such as job satisfaction, performance, turnover intention, and role stress. Hence, it is unclear whether WFH affects employee wellbeing positively or negatively, as the evidence from the existing literature is indeterminate and often contradictory.

Since the outbreak of the COVID-19 pandemic, the concept of WFH has gained increased attention and additional meaning, as the proportion of people who worked from home experienced an unprecedented increase during the pandemic. For instance, Bick and colleagues [6] found that 35.2 percent of the workforce in the United States worked entirely from home in May 2020, up from 8.2 percent in February 2020. In addition, for many employees, the change in the overall work situation has been sudden and mandatory, leaving them unprepared for the transition [7]. Thus, the existing literature on WFH may not be applicable to the pandemic situation [8]. Our study aimed to contribute to the literature on WFH during the COVID-19 pandemic in the following three ways. First, as previous studies have conducted some descriptive analyses, such as the demographic characteristics of WFH employees after the outbreak of the pandemic [6,9], more empirical studies are needed to further examine how theoretical models such as the Job Demands–Resources (JD-R) model works in the WFH context during the COVID-19 pandemic. Second, although previous studies compared employees WFH and employees going to work, accurate comparisons were difficult, and the results are inconsistent, as groups of employees WFH are usually much smaller than those of in-office employees [1]. Thus, lockdown policies, including the transition to WFH during the COVID-19 pandemic, offer a unique opportunity to explore differences in experience between employees who worked from home and those who still went to work. Third, different countries responded differently to the COVID-19 pandemic, and few studies have contributed to the literature on WFH during the pandemic in a Norwegian context. In Norway, a national lockdown was announced on 12 March 2020 [10], and since then, working from home has been the main policy of many organizations. Nordic countries, such as Norway, have dealt with the COVID-19 crisis more efficiently than most other countries [11] due to the ease with which citizens maintain social distance and thus prevent spreading the virus. Norway also has a public welfare system that is well adapted to reduce the negative influence of redundancies, unemployment insurance, and sick leave.

Therefore, in this study, based on data collected in Norway in February 2021, we drew from the Job Demand–Resources model [12,13] as well as the Conservation of Resources (COR, [14,15]) theory and aimed to explore how relationships between job demands (work–home conflict and workload), job resources (coworker support, leader support, and family–work facilitation), and wellbeing (burnout and work engagement) have been experienced by employees who changed from working in the workplace before the pandemic to working from home during the pandemic (change group) and those who remained working in their workplaces before and during the COVID-19 pandemic (no-change group).

### 1.1. The COVID-19 Pandemic, Work Situation, and Wellbeing

The pandemic has been proved to increase mental health risks to a large extent. For instance, a longitudinal study from the UK reported that the percentage of individuals with an anxiety disorder almost doubled during COVID-19 at 24% (95% CI 23%, 26%) compared to the pre-pandemic level of 13% (95% CI 12%, 14%) [16]. Research has also illustrated an everyday life characterized by loneliness and poorer mental health in the COVID-19 context [17]. In Germany, researchers have demonstrated that the lockdown circumstances generally have a negative effect on the satisfaction with the work and home life of individuals [18].

Some researchers have already made great efforts to investigate the influence of WFH during the pandemic on employees’ mental health and work-related attitudes or behaviors. According to Carillo et al. [7], epidemic-induced WFH inherits some of the characteristics from conventional WFH, but it appears to have particular aspects that make it a unique context with specific conceptual boundaries. First, it became more of a mandatory requirement than a voluntary option. Individuals across occupations who preferred going to work were forced into WFH [19]. Second, it occurred not only in certain businesses that may have a preference for WFH and whose employees are more productive if they are WFH, such as internet companies, but also in other traditional occupations (e.g., teachers in education). Thus, employees all over the world have worked substantially more from home, with their social skills in greater demand, being forced to communicate with their colleagues and clients remotely due to restrictions [20]. Further, employees may face challenges due to fundamental issues such as lacking space in one’s home to attend to work and having difficulty receiving instant or efficient feedback from students or colleagues [21]. Research on WFH during the pandemic has revealed mixed results. Employees who worked from home had a high sense of insecurity, increased work–home conflict, and high levels of stress [22,23]. This was further supported by Hayes et al. [24], who indicated a higher level of burnout and stress among US employees WFH compared to before the pandemic when they went to the workplace. It was also reported that WFH was associated with higher job performance due to its negative association with stress in a Canadian population [23]. Moreover, Dubey and Tripathi [25] analyzed Twitter activity and found that the WFH concept was viewed positively by employees. Thus, researchers and practitioners are concerned with the influence of the workplace change on employee wellbeing during the COVID-19 pandemic [24]. Meanwhile, for employees who remained working on the frontline under the pandemic, such as health care workers, virus-related stressors such as the fear of being infected as well as the fear of infecting their family have also become major concerns, which may be detrimental to their wellbeing [26,27,28].

Despite these findings, few studies have investigated both employees who worked from home and those who remained in the workplace. In the pandemic context, a qualitative study [8] identified four key remote work challenges (work–home interference, ineffective communication, procrastination, and loneliness), as well as four virtual work characteristics that affected the experience of these challenges (social support, job autonomy, monitoring, and workload). Therefore, work–home conflict and workload can be identified as important demands that are highly related to WFH. In addition, social support appears to be a beneficial resource for employees under a lockdown situation when they may face the challenge of loneliness. In particular, employees may have different perceptions about different sources of social support (e.g., leaders, coworkers, and family support) under such circumstances. Furthermore, burnout and engagement have been considered important indicators of employee wellbeing and have been shown to be highly related to work-related attitudes and behaviors such as job satisfaction, commitment, and performance [29,30].

In our study, we aimed to explore the importance of workload, work–home conflict (WHC), and social support, including leader support, coworker support, and family–work facilitation (FWF), in relation to wellbeing (burnout and work engagement) among employees who still went to the workplace (no-change group) and employees who transitioned to WFH (change group) during the COVID-19 pandemic. At the time of the data collection, Norway was working to curb the second wave of infections and was under lockdown to prevent overloading the healthcare system. Thus, employees in these two groups may have experienced their work situations differently during the period of study. Considering the previous literature indicating that employees in both the change and no-change group may encounter different stressors under the pandemic, we propose the following research question:


**Research question 1 (RQ1).**
*Are there significant differences in levels of WHC, workload, coworker support, leader support, and FWF between employees in the change and no-change group?*


### 1.2. Theoretical Framework

Given the variables that we included in our study, we combined the JD-R model and the COR theory to explain our study hypothesis. According to the JD-R model, all types of job characteristics can be classified into one of two categories: job demands and job resources [12,13]. Job demands are defined as physical, psychological, social, or organizational aspects of the job that require sustained physical and/or psychological effort and are therefore associated with certain physiological and/or psychological costs [13]. On the other hand, job resources refer to physical, psychological, social, or organizational aspects of the job that are necessary to achieve work goals, reduce job demands and the associated physiological and psychological costs, or stimulate personal growth, learning, and development [13]. Burnout is traditionally characterized as a syndrome of exhaustion, cynicism, and lack of efficacy experienced by employees [31,32,33]. In contrast to burnout, engagement is a positive, fulfilling, work-related state of mind characterized by vigor, dedication, and absorption [34]. Numerous researchers have reported that job demands are unique predictors of exhaustion, whereas job resources are unique predictors of engagement [29,34,35,36,37].

In addition, in this study, the COR theory may serve to explain the importance of resources outside of work, such as family–work facilitation. According to this theory, humans are driven to foster, create, preserve, and protect their resources [38]. A central principle in the theory is the gain spiral, where individuals with greater resources find it easier to obtain new resources. In addition, people who possess resources do encounter stressful situations, but they are better equipped to deal with these stressors [38].

### 1.3. Relationship between Job Demands/Job Resources and Burnout/Work Engagement in the Context of WFH

As is widely supported by the literature and the JD-R model, WHC has been an important issue related to WFH [3]. As the current pandemic continues to unfold, the potential for conflict between work and home spheres may be greater than ever [39]. Some researchers have focused on the effect of WHC on productivity during the COVID-19 pandemic [40], but more empirical studies are still needed to further investigate the influence of WHC during the pandemic from different perspectives. For instance, comparing the relationship between WHC and employee wellbeing among the change and no-change group may better reflect the overall influence of WHC during the pandemic.

Moreover, researchers have revealed that workload has been a noteworthy job demand during the pandemic, as some employees stated that their work seemed never-ending when working from home [8]. Some employees also believed that people could decide to work right away or procrastinate while working from home, which was associated with their workload. Sadiq [41] found that WHC mediated the relationships of workload with job stress and job dissatisfaction in Pakistani police. Based on the above, we consider WHC and workload to be important job demands that may influence employees’ burnout and work engagement. In sum, we intended to test the following hypotheses among both the change and no-change group:

**Hypothesis** **1** **(H1).**
*WHC is (a) positively related to burnout and (b) negatively related to work engagement.*


**Hypothesis** **2** **(H2).**
*Workload is (a) positively related to burnout and (b) negatively related to work engagement.*


When it comes to job resources, researchers have largely determined that social support plays a beneficial role for employees who work from home. That is, if employees receive more social support, they will perform better when working from home. For example, Wang et al. [8] utilized quantitative data to prove that social support had a positive effect on performance due to less procrastination and home–work interference; a negative effect on emotional exhaustion due to less procrastination, work–home interference, and loneliness; and a positive effect on life satisfaction due to less work–home interference and loneliness. Szkody et al. [42] also indicated that social support buffers the connection between worry about COVID-19 and psychological health. Moreover, Xiao et al. [43] proved that social support provided to medical staff caused a reduction in anxiety and stress levels and increased their self-efficacy. Data from Thailand also showed that supervisor support had a negative effect on employees’ emotional exhaustion [44].

However, the beneficial effect of social support has been challenged, as Deelstra et al. [45] proved that the reception of support evokes feelings of incapacity, which in turn evokes poor self-image in employees. Furthermore, the literature highlights that during the COVID-19 pandemic, social support has changed to digital support, and consequently, it is unclear how effective support is in improving employees’ state of wellbeing [8,39]. In this study, we included three different sources of social support, namely, leader support and coworker support as job resources and FWF (family support) as a home resource, to comprehensively examine the effects of different kinds of social support on burnout and work engagement.

In line with the COR theory, workers who experience high FWF will be in a better position to invest these resources in their behavior and attitudes to deal with demanding situations and foster other resources. For example, workers who see their family as benefiting from their work will consequently be more energized and enthusiastic toward their work [46] and, thus, be more engaged. Furthermore, we are interested in how these three sources of social support differ in relation to employee wellbeing to see which is more important for employees in the change and no-change group. Therefore, we propose the following hypotheses for both the change and no-change group:

**Hypothesis** **3** **(H3).**
*Leader support is (a) negatively related to burnout and (b) positively related to work engagement.*


**Hypothesis** **4** **(H4).**
*Coworker support is (a) negatively related to burnout and (b) positively related to work engagement.*


**Hypothesis** **5** **(H5).**
*FWF is (a) negatively related to burnout and (b) positively related to work engagement.*


## 2. Materials and Methods

### 2.1. Study Design

The present study was based on a larger student project named “Healthy workplaces in light of COVID-19” using a questionnaire of previously validated scales related to wellbeing at work. The online survey was conducted using the survey service “Nettskjema”, provided by the University of Oslo in Norway. The invitation, consent information, and the link to participate in the study were distributed by undergraduate students via email. Employees working in (1) scientific, technical, and administrative services and (2) health and social services were invited to participate in the study. These occupational groups were selected to collect data from employees most likely to work from home during lockdown (scientific, technical, and administrative services) and from employees likely to go to work during the pandemic (health and social services). Convenience (acquaintances) and snowball sampling methods were implemented between 25 January 2021 and 7 February 2021. This was during a period where the level of lockdown in Norway would indicate that most employees were working from home if they had the opportunity to do so. The study was conducted in accordance with the guidelines of the Norwegian Center for Research Data (NSD).

### 2.2. Instruments

All the scales were measured on a 5-point Likert scale, with responses ranging from never to always (N-A) or from totally disagree to totally agree (D-A). In addition to the variables applied in this study, the survey also included demographic measures such as gender and age.

Utrecht Work Engagement Scale-3 (UWES, [47]) is a shortened version measuring the employee experience of being engaged at work with 3 items (a = 0.78). Each of the 3 items was meant to capture one of the subdimensions of work engagement—absorption, vigor, and dedication—on a scale ranging from N to A. One example is, “At my work, I feel bursting with energy”.

Burnout Assessment Tool (BAT, [48,49]) measures the employee experience of being burned out with 12 items (a = 0.88). BAT is composed of 4 subdimensions, including exhaustion, mental distance, and emotional and cognitive impairment, each measured with 3 items. All the items ranged from N to A. An example is: “At work, I feel mentally exhausted”.

Family-to-Work Facilitation (FWF, [50]) measures the level of positive spillover from family to work and consists of 3 items (a = 0.77) scored from D to A. One example is, “My family helps me acquire skills and this helps me be a better worker”.

Work–Home Conflict (WHC, [51,52]) measures the level of negative spillover from work to family and consists of 4 items (a = 0.84) ranging from D to A. One example is, “Stress at work makes me irritable at home”.

The last three scales were obtained from “Questionnaire on the Experience and Evaluation of Works” [53] and were all scored from N to A: (1) Social Support Coworker (SSC) measures social support from coworkers with 3 items (a = 0.72), such as: Can you count on your colleagues for help and support, when needed?” (2) Social Support Leader (SSL) measures social support from managers with 3 items (a = 0.87), such as: “Do you feel your work is recognized and ﻿appreciated by your supervisor?” Finally, (3) Workload (WL) measures the experience of work overload with 4 items (a = 0.77), for example: “Do you have too much work to do?”

Control Variables. Age and gender were included as control variables. Age was divided into 5 categories: (1) <31, (2) 31–40, (3) 41–50, (4) 51–60, and (5) <60. Values for Gender were 1 for females and 2 for males.

Work Situation. Lastly, measures about (1) current and (2) pre-COVID-19 (a year earlier) work situations were included, asking “Which alternative gives the best description of your work situation the last month (current)/before COVID-19 (pre-COVID-19)?” Both were scored on a 3-point scale with the alternatives: (1) I am (was) working from home, (2) I (used to) work both from home and at the workplace, and (3) I go (went) to work every day. The two groups studied, the no-change and change group, were created based on the participants’ combination of current and pre-COVID-19 work situation.

### 2.3. Sample

In total, 629 participants completed the survey. Out of the total sample, 575 participants met the criteria for inclusion in this study as either belonging to the change or no-change group. Three hundred and eighty-three of these participants were women (66.6%), and most of the respondents were in the age groups <31 (27.8%), 41–50 (19.8%), and 51–60 (32.7%). The sample consisted of employees representing health and social services (43.5%), professional, scientific, and technical services (48.0%), and other (8.6%). A large proportion of the participants reported having completed a higher education of 3 years or more (78.8%). Three hundred and ten participants were living with children (53.9%), and four hundred and four lived with a partner (70.3%).

The no-change group consisted of 269 participants who went to work every day both pre-COVID-19 and during the COVID-19 pandemic; hence, their work situation did not change due to the pandemic. The change group was composed of 306 participants who changed their work situation from going to workplaces before the pandemic to working more from home during the pandemic (see Table 1). In the no-change group, 16 percent were employed in professional, scientific, and technical services, and 75 percent were in health and social services. Conversely, in the change group, 76 percent were employed in professional, scientific, and technical services, and 16 percent were in health and social services.

### 2.4. Data Analysis

Descriptive analysis was conducted in IBM SPSS Statistics for Macintosh, version 27 (Armonk, NY, USA), to determine the central tendency and dispersion of the sample, the mean and standard deviation for continuous variables, and frequencies for categorical measures. In addition, the Pearson product–moment correlation coefficient (Pearson’s r) was used to measure the bivariate correlations between study variables. The group means of the study variables were then compared through 7 independent sample *T*-tests with false discovery rate (FDR) 0.05 *p* value cut-off correction. FDR is a procedure that provides the opportunity to control for the probability of Type I errors while, unlike the Bonferroni correction, allowing a reduction in Type II error [54].

The relationships between the predictors and the outcomes were investigated by multivariate structural equation modeling (SEM) in MPLUS version 8.4 [55]. As the model test statistics of SEM models (*χ*^2^) are sensitive to sample size, it is recommended to also consider the Standardized Root Mean Square Residual (SRMR) and three approximate fit indices, namely, Steiger–Lind Root Mean Square Error of Approximation (RMSEA), the Bentler Comparative Fit Index (CFI), and the Tucker–Lewis Index (TLI), to evaluate the model’s goodness of fit [56]. According to Hooper et al. [57], CFI and TLI values ﻿>0.90 and SRMR and RMSEA values < 0.08 indicate an acceptable fit. As all of the variables applied in this study were measured using a self-report questionnaire, we followed the recommendations of Podsakoff et al. [58] to address limitations due to common method bias, evaluating the goodness of fit of the one-factor model compared to that of the study model.

## 3. Results

### 3.1. Assumptions for T-Test and Structural Equation Model

The assumptions of normality and homogeneity were considered before conducting the *T*-test. Skewness and kurtosis levels were below the recommended threshold of 1 for all the variables in both groups. Levene’s test was significant for one variable; thus, the corrected test results were reported for leader support. Before conducting the SEM analysis, we tested for the violation of the assumptions of linearity and multicollinearity. The curve estimation for all the relationships in both groups was sufficient for covariance-based SEM. Additionally, the multicollinearity statistics indicated that the assumption was not violated, mean VIF = 1.298.

### 3.2. Descriptive Statistics and Independent Sample T-Test

The bivariate correlations between the study variables for the two groups (change and no-change) are displayed in Table 2. The strongest correlation was between burnout and WHC in the no-change group and between coworker and leader support in the change group. The descriptive statistics of the study variables are displayed in Table 3. The highest mean value was found for coworker support in the no-change group, whereas work engagement presented the highest mean value in the change group. In both the no-change and change group, the lowest mean score was found for burnout.

The results of the independent sample *T*-tests evaluating the mean differences in the study variables between the no-change and change group are also presented in Table 3. The no-change group (*M* = 4.03, *SD* = 0.58) experienced significantly more coworker support (*t* (573) = 2.534, *p* < 0.05) than the change group (*M* = 3.90, *SD* = 0.64). In addition, the change group (*M* = 3.53, *SD* = 0.60) experienced a higher workload (*t* (573) = −2.989, *p* < 0.05) than the no-change group (*M* = 3.37, *SD* = 0.65). No significant results were found for the other variables.

### 3.3. Test for CFA and Common Method Bias

The model fit indices for the CFA of the study model indicated an acceptable model fit when the four subdimensions of BAT were included (*N* = 575, *χ*^2^ (439) = 1005.22, *p* < 0.05; CFI = 0.920, TLI = 0.910, SRMR = 0.053, RMSEA = 0.047, 90% CI [0.044, 0.051]). When testing for common method bias, the one-factor model did not provide acceptable fit indices (*N* = 575, *χ*^2^ (418) = 1652.90, *p* < 0.05; CFI = 0.825, TLI = 0.793, SRMR = 0.435, RMSEA = 0.072, 90% CI [0.068, 0.075]), and hence, common method bias did not seem to be an issue. The control variables, age and gender, were not included in the measurement model.

### 3.4. The Structural Equation Model

The results of the analysis of the SEM model for work engagement and burnout in the change and no-change group are presented in Table 4 as well as Figure 1 (no-change group) and Figure 2 (change group). The results provide support for hypothesis 1; WHC was positively related to (a) burnout in both the no-change (*b* = 0.43,* p* < 0.001) and change group (*b* = 0.36,* p* < 0.001) and negatively related to (b) work engagement in both the no-change (*b* = −0.19,* p* < 0.01) and change group (*b* = −0.26,* p* > 0.001).

The results provide partial support for hypothesis 2 in both the no-change and change group; workload was positively related to (a) burnout only in the no-change group (*b* = 0.14, *p* < 0.05), whereas for the change group, workload was positively, not negatively, related to (b) work engagement (*b* = 0.20, *p* < 0.01).

Testing hypotheses 3, 4, and 5 regarding job resources, leader support, coworker support, and FWF also provided mixed results. Hypothesis 3 was partly supported, as leader support was negatively related to (a) burnout only in the change group (*b* = −0.17,* p* < 0.05).

Further, hypothesis 4 was also partly supported for both groups, as coworker support was positively related to (b) work engagement in both the change (*b* = 0.20, *p* < 0.05) and no-change group (*b* = 0.19, *p* < 0.01).

Lastly, the results provided support for hypothesis 5a, as FWF was negatively related to burnout only in the change group (*b* = −0.15, *p* < 0.01). Hypothesis 5b was supported in both groups, as FWF was positively related to work engagement in the no-change (*b* = 0.15, *p* < 0.05) and change group (*b* = 0.26, *p* < 0.001).

## 4. Discussion

The aim of this study was two-fold: (1) to explore differences in the experience of job demands and resources between employees who still went to their workplaces (no-change group) and employees who transitioned to WFH (change group) during the pandemic; (2) to explore the relationships among job demands, job resources, and wellbeing outcomes for employees in these two groups. Our results indicate that the work situation may have deteriorated somewhat for employees in the change group, as they experienced less support from their coworkers and a higher workload than employees in the no-change group (RQ1). There appeared to be no significant difference in other demands and resources. One explanation for this result is that at the time of data collection in this study, employees in Norway had been experiencing lockdown policies for almost one year. As a result, employees in the change group may have adapted to this “new normal”. The other reason may be due to the unique context in Norway. Due to protection from the Norwegian Working Environment Act, employees may suffer less from employment issues caused by the pandemic, such as unemployment, sick leave, and health risks. Employers are obligated to ensure that work does not negatively affect employees’ physical and mental health. That is, despite the shock of the COVID-19 pandemic, employees can receive support from their employers to ensure that they have fewer job demands and more job resources, which can prevent the development of negative work-related outcomes. The third reason could be that employees in the change and no-change group may experience distinct stressors. As we mentioned before, employees in the change group may face the possible challenges of WFH while employees in the no-change group may encounter the fear of infection as they still must keep close contact with the public. Employees in the two groups may all have experienced work stress during the pandemic.

Moreover, we argued that it was necessary to further test the relationship between job demands and resources and burnout and engagement in a Nordic context during the COVID-19 pandemic. To the best of our knowledge, this study is the first to test these relationships separately for employees in the change and no-change group. Regarding job demands, the relationship between WHC (H1) and wellbeing was mostly detrimental in both groups, increasing burnout and reducing work engagement. In addition, workload (H2) was associated with increased burnout solely in the no-change group but was positively related to more work engagement in the change group. When looking at job resources, leader support (H3) was an important factor and was negatively related to burnout in the change group, whereas coworker support (H4) was positively related to work engagement in both groups. Lastly, FWF (H5) seemed to be the most important source of support for the change group, as it appeared to relate to both burnout and work engagement in a positive way. FWF was also positively associated with work engagement in the no-change group.

We further summarize the results of our study in the following points: First, our study reflects the JD-R model on the relationships between job demands and burnout as well as job demands and work engagement. Based on previous literature [8], we believed that WHC and workload were two noteworthy job demands for employees who worked from home during the COVID-19 pandemic. Our results support previous studies [26,39,59] that indicate that WHC could be the most influential factor related to burnout and work engagement, with relatively high coefficients in both the change and no-change group. As for workload, for employees in the change group, a greater workload was not related to more burnout but was related to more work engagement, which was a surprising finding in this study. For employees in the no-change group, a higher workload was positively associated with more burnout, as expected. One reason that may explain the unexpected finding is that workload could be considered a challenging demand [60,61,62] for employees who work more from home. A higher workload can motivate and encourage employees to concentrate on work in a somewhat distracting context. Furthermore, our study confirmed that challenging demands may be experienced as hindering demands (and vice versa) depending on the context [63], as employees in the two groups perceived workload differently. The other reason may be related to the characteristics of the sample in the change group, as 76 percent of them were in professional, scientific, and technological occupations, whereas 75 percent of participants in the no-change group were healthcare and social workers. The characteristics of workload are quite different in these two groups; that is, a higher workload in the change group may entail more teaching or writing, whereas a higher workload in the no-change group may mean more patients or clients. Therefore, employees in the two studied groups may have perceived their workloads differently. In addition, the reason why workload was not significantly related to burnout for the change group but significantly associated with burnout in the no-change group could mean that for employees in the change group, WFH may have some advantages such as providing a more comfortable environment during the lockdown, more freedom of action, and reducing the time and costs of commuting. Therefore, the negative relationship between workload and burnout may be buffered by the advantages of WFH for employees in the change group.

Second, in our study we discovered different relationships between the three sources of social support and wellbeing for the two groups of employees. Generally, our study proved that social support had different relationships with burnout and engagement for the change and no-change group. Specifically, for employees in the change group, leader support seemed to be more important for reducing burnout, as it was negatively related to burnout. Coworker support seemed to be more beneficial for promoting work engagement, as it was positively related to engagement. Furthermore, FWF was significantly related to both burnout and work engagement in the change group. Therefore, for employees who transitioned to more WFH, family support seemed to be a vital element to prevent burnout and promote work engagement. For employees in the no-change group, coworker support and FWF were both positively related to work engagement but showed no significant relationship with burnout. Additionally, leader support had no significant relationship with either burnout or engagement in the no change group. Overall, these results, to some degree, indicate that social support had a stronger relationship with work engagement, but neither were related to burnout in the no-change group. However, in the change group, social support seemed to be related to both work engagement and burnout.

### 4.1. Limitations

Our study contributes to the literature on working from home during the pandemic by investigating the relationships among demands, resources, and employee wellbeing in the change and no-change group. However, there are some limitations that should be noted. First, we utilized snowball sampling to collect our data, which is a nonprobability sampling technique where subjects are recruited through acquaintances. This approach provided convenience for our data collection but also led to nonrandom sampling, which might undermine the representativeness and generalization of the results. In addition, the occupations of employees in the change and no-change group in this study were not equally distributed. The distribution bias of our sample might influence the generalization of our results. To some degree, our sample reflected the reality of occupations that had a higher probability of working from home or remaining in the workplaces during the COVID-19 pandemic. However, the sample did not cover employees with a lower socioeconomic status, such as blue-collar workers, who had to work on the frontline during the pandemic. There have also been large and marked differences for COVID-19 health personnel in Norway compared with other countries. Among other factors, the pressure on health personnel may have been reduced due to the infection being controlled effectively: in 2020, Norway had one of the lowest death rates, with a total of 228 deaths among over 5 million inhabitants [64]. Therefore, even though the sample in this study was not equally distributed, we believe that the results may still have some implications for the future of work.

Second, our study used a cross-sectional research design, which was insufficient for drawing conclusions about causal relationships between variables. Future research may consider using a longitudinal design to test the causal relationships among job demands, job resources, burnout, and work engagement within the context of the COVID-19 pandemic.

Third, our study was based on self-report measures, which may lead to common method bias. Even though we tested for common method bias, and it was not an issue in this study, multiple methods of assessment should be considered in future studies. For example, future studies could consider utilizing objective measurements to evaluate employees’ workload.

### 4.2. Implications

Our study indicated that there was a difference in the experience of workload and coworker support between the change and no-change group. Moreover, even though we found that all of the investigated job demands and resources were significantly related to employee wellbeing to some extent, there were some differences between the two groups regarding the importance of different job demands and job resources for employee wellbeing. Thus, practitioners working to increase wellbeing for both employees in the change and no-change group should focus their interventions on facilitating a healthy work–home balance, as WHC was the most important factor related to employee wellbeing in both groups of this study. This may have additional importance for employees WFH, as support from family was found to be the most important resource related to wellbeing. In addition, to reduce strain, interventions should focus on leader support, whereas to facilitate motivation, coworker support is important for the wellbeing of employees working from home. Thus, facilitating access to multiple sources of support and family-friendly policies seems to be an important focus area of healthy workplace interventions for employees in both working conditions.

When it comes to the association between workload and wellbeing, the results from this study may be of particular interest to both researchers and practitioners. For employees still at work, workload seems to contribute to strain. However, even though employees WFH reported a higher workload, it seems to have motivational effects for them. On the one hand, this is an important finding for practitioners and managers occupied with the wellbeing of employees working in different work situations. On the other hand, this contributes to research on the JD-R model by demonstrating that the same job demand can be experienced as hindering or challenging for employees depending on their work situation. Moreover, it seems that for the different groups, different job resources are of importance for different outcomes. Therefore, practitioners should be aware of this when seeking to affect either motivational or health-impairing processes related to employee wellbeing. Researchers should also be aware of these mechanisms, especially when generalizing findings across work situations.

## 5. Conclusions

The COVID-19 pandemic swept across the world at the beginning of 2020 and fueled the fire of remote work. The aim of this study was to contribute to the literature on working from home during the COVID-19 pandemic. Employees who transitioned to working more from home and those who still went to work were studied through the framework of the job demands–resources model and the conservation of resources theory in Norway, a country praised for its handling of the pandemic. The results of this study indicate that work–home conflict was an important job demand that was related to employee wellbeing in both groups. Workload showed different associations with wellbeing in the two groups. In addition, the three different sources of social support, namely, leaders, coworkers, and family, presented a positive relationship with employee wellbeing in different ways. Moreover, family support appeared to be more strongly related to wellbeing for employees in the change group. Therefore, employers may have a lot to gain from facilitating for family-friendly policies and social support from different sources in order to both reduce employee health impairment and increase their motivation.

## Figures and Tables

**Figure 1 ijerph-19-01982-f001:**
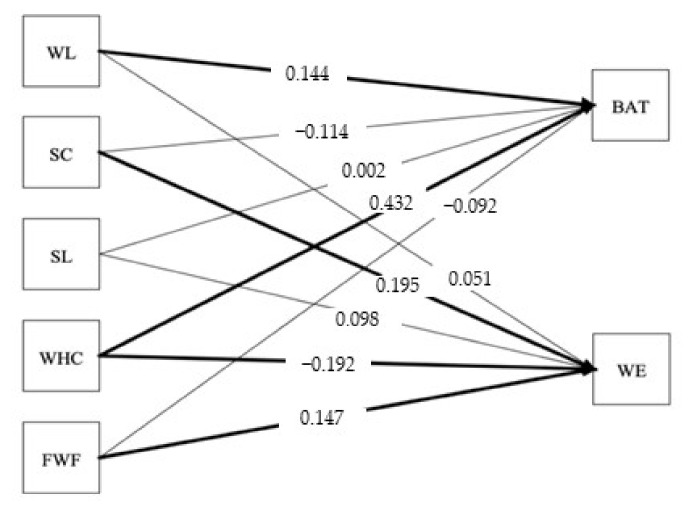
The proposed structural model with standardized coefficients for the no-change group. Notes: WL = Workload, SC = Coworker Support, SL = Leader Support, WHC = Work–Home Conflict, FWF = Family-to-Work Facilitation, BAT = Burnout, WE = Work Engagement; the thicker lines mean significant relationships, *p* < 0.05.

**Figure 2 ijerph-19-01982-f002:**
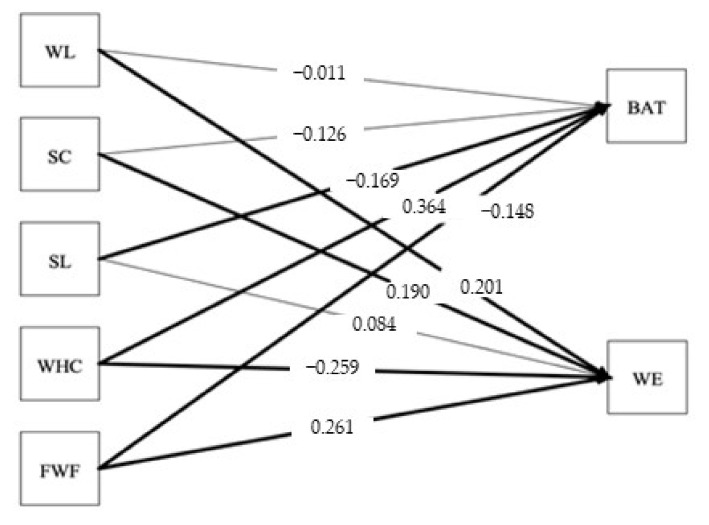
The proposed structural model with standardized coefficients for the change group. Notes: WL = Workload, SC = Coworker Support, SL = Leader Support, WHC = Work–Home Conflict, FWF = Family-to-Work Facilitation, BAT = Burnout, WE = Work Engagement; the thicker lines mean significant relationships, *p* < 0.05.

**Table 1 ijerph-19-01982-t001:** The distribution of participants grouped by their work situation before and during the pandemic.

		During	
Before	Work from Home	Workplace and Home	Work from Workplace
Work from Home	-	-	-
Workplace and Home	**40 (6.4%)**	-	-
Work from Workplace	**155 (33.4%)**	**111 (17.7%)**	269 (42.8%)

Notes: The no-change group is in shadow; the change group is in bold.

**Table 2 ijerph-19-01982-t002:** Bivariate correlations of the study variables for the no-change group (N = 269) and change group (N = 306).

	1	2	3	4	5	6	7
No-Change group							
1. Work Engagement	-						
2. Burnout	−0.42 **	-					
3. FWF	0.22 **	−0.11	-				
4. Coworker Support	0.34 **	−0.28 **	0.23 **	-			
5. Leader Support	0.27 **	−0.20 **	0.13 *	0.46 **	-		
6. WHC	−0.27 **	0.55 **	−0.01	−0.30 **	−0.29 **	-	
7. Workload	0.00	0.29 **	0.07	−0.08	−0.04	0.40 **	-
Change group							
1. Work Engagement	-						
2. Burnout	−0.45 **	-					
3. FWF	0.31 **	−0.15 **	-				
4. Coworker Support	0.34 **	−0.32 **	0.14 *	-			
5. Leader Support	0.27 **	−0.35 **	0.05	0.56 **	-		
6. WHC	−0.28 **	0.48 **	−0.03	−0.29 **	−0.27 **	-	
7. Workload	0.11 *	0.13 *	0.11	−0.00	−0.03	0.42 **	-

Notes: FWF = family-to-work facilitation; WHC = work–home conflict. Pearson’s correlation coefficients. ** *p* < 0.01, * *p* < 0.05.

**Table 3 ijerph-19-01982-t003:** Independent group *T*-test on study variables between the no-change and change group.

**Variables**	**No-Change Group**		**Change Group**		
	*M*	*SD*	*M*	*SD*	*T*-test
FWF	3.56	0.69	3.60	0.63	−0.742 ^ns^
Coworker Support	4.03	0.58	3.90	0.64	2.534 *
Leader Support	3.76	0.91	3.87	0.81	−1.568 ^ns^
WHC	2.92	0.89	2.93	0.85	−0.073 ^ns^
Workload	3.37	0.65	3.53	0.60	−2.989 *

Notes: *M* = Mean. *SD* = Standard Deviation. ^ns^ = non-significant FDR 0.05 *p* value corrected. * *T*-test significant at FDR 0.05 *p* value corrected.

**Table 4 ijerph-19-01982-t004:** Standardized coefficients and *R*^2^ of the structural model.

Variables	No-Change Group		Change Group	
	Burnout	Engagement	Burnout	Engagement
	*β*	*β*	*β*	*β*
Gender	0.02	−0.10	−0.01	0.05
Age	−0.18 ***	0.16 **	−0.25 ***	0.11 *
WHC	0.43 ***	−0.19 **	0.36 ***	−0.26 ***
Workload	0.14 *	0.05	−0.01	0.20 **
Leader support	0.00	0.10	−0.17 *	0.08
Coworker support	−0.11	0.20 *	−0.13	0.19 **
FWF	−0.09	0.15 *	−0.15 **	0.26 ***
*R* ^2^	0.36	0.22	0.36	0.27

Notes: *** *p* < 0.001, ** *p* < 0.01, * *p* < 0.05.

## Data Availability

The data presented in this study are available on request from the corresponding author. The data are not publicly available due to privacy.

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
