# Peer review of "To Change or Not to Change: A Study of Workplace Change during the COVID-19 Pandemic"

_ijerph, 2022, doi:10.3390/ijerph19041982_

Round 1
Reviewer 1 Report
The authors conducted a very useful study of the psychological effects of changes in the workplace. Interesting data have been obtained, which may also be of practical importance for the management of activities in various fields of work. Therefore, the article should be published. Meanwhile, there are several proposals, in our opinion, that can improve the text for the reader's perception. It seems to me that working "at home" has a number of advantages in a number of industries (for example, IT, accounting, literature, etc., for teachers too). Therefore, it is not entirely correct to compare the burnout of those who have direct contact with others and those who have indirect contact with others. An equally important point is the place of residence while working at home: an apartment or a detached house are very different things for a comfortable stay during a strict quarantine. In addition, people have not been working "at home" for so long to catch the negative sides of this phenomenon (for example, some considered such work as very convenient in view of the lack of need to move around the city by transport, more freedom of action, etc.). Perhaps that is why those who work "at home" did not find a connection with burnout. Secondly, it would be good to reflect in the text which hypotheses are tested in separate paragraphs of the "Results". It would also be useful to present SEM models in the form of figures with indicators of relationships and regression coefficients. Finally, it would be desirable to provide a basis for applying the parametric method of comparison and modeling. Have the data been checked for the normality of the distribution of variables in each sample (the samples are not equalized)?
Author Response
Please find our response to your comments in the attached word file.
Best regards,
Silje Fladmark

Reviewer 2 Report
To Change or not to Change: A Study of Workplace Change During the COVID-19 Pandemic
This is a novel study with timely data gathered during this historic global pandemic. The authors’ research questions regarding the effects on employee well-being of having to “work from home” (WFH) versus continuing to work at a workplace during the pandemic are important. They apply two theoretical models to these questions including the job demands-resources models and a social support model (COR). The hypotheses, based on these models, are clear and testable. However, as a cross-sectional study based on a convenience sample, it is not ideal to show causality as the author’s state. More concerning is that the sample may not represent the working population of Norway or other countries outside Europe as it is disproportionately from two sectors – health care and social services, and professional, scientific, technical employees. There are no lower SES workers (blue collar workers who may have to work on the front lines of the pandemic under already challenging working conditions). Inclusion of these workers may change their results significantly regarding workload and work-family conflict since even in Norway low-income jobs do not have the same resources available as these higher income occupations. However, regardless of the limitations of this study, they do have some significant findings regarding job demands, support and employee health and well-being, although in the end there were limited differences between the two groups.
Introduction/Background
The introduction and review sections are clear and well-structured. They have included relevant references to the JDR model, burnout and the recent and past literature on working from home. I would suggest revising the Abstract to make it clearer that the “change” and “no change” groups refer to “mandatory working from home” and “continue work at a workplace” during the pandemic. It is not clear.
Research Design
The research design, despite its stated limitations, is sound. However, it is necessary to clarify Table 1 somewhere in the Introduction or Methods e.g. how and why people were grouped into the “No Change” or “Change” group, since there were a couple of people (n = 2) who were categorized in the “No Change” group who went from working from home before the pandemic to a “hybrid” arrangement – working from home and the workplace both. These people could just as well be categorized as a “change”? It is also difficult to see the “Italics” in Table 1 that are supposed to designate the “no change” group.
The other design concern is regarding the sample. Instead of measuring “working from home” or “changing from one work arrangement to another” what we may be seeing is differences in the type of work during the pandemic. Most (75%) of those who remained working at a workplace were health care workers, while most (76%) who changed to “working from home” were “Professional, Scientific, Technical” employees. The authors need to make this flaw clearer in the Methods section and to justify it somehow.
Also, there while there is some study of “working from home” and its effects, it is not clear why the authors believe “changing” work locations versus “not changing” and remaining at the workplace during a pandemic will have an effect on employee health and well-being? Some could argue that for workers who remain on the frontlines, the fear of the virus, close contact with the public, could cause more stress than being “required” to work from home. This should be stated more clearly in the introduction.
Ethics – I do not observe any ethical concerns.
Methods
The sample size of n = 575 is sufficient to analyze differences between the two groups. I am concerned about the effect of possible misclassification. There are a small number of workers n=2 who “worked from home” prior to the pandemic and then “worked from home and at the workplace” during the pandemic who are classified as “no change”. While it is a small number, these could also be classified as “changing and working from home”. This needs to be fixed. Either omit them, or explain which group they should be in.
Results/Discussion
The Tables (except for Table 1) were clear and easy to understand.
Most of the hypotheses were supported or partially supported. The “Work from Home/Change” group reported significantly higher workload and lower coworker support than the “Workplace/No Change” group. However, the WFH group with higher workloads experienced greater engagement whereas those in a workplace experienced high workload and higher burnout. The authors suggest this unusual finding may be due to differences in the occupations between groups. Health care workers workload may be due to more patients whereas the workers at home may perceive a high workload as a way of preventing distraction while at home, which makes sense. The Work-Home Conflict increased burnout and lowered engagement in both groups. Leadership support decreased burnout in those working from home and coworker support increased engagement for both groups. Family support also helped increase engagement in both groups, but it also lowered burnout just for those working from home. Overall, the discussion raised more questions about the occupational difference between the groups but overall the discussion was more than adequate.
Conclusion
The conclusion that support was more important for the Work from Home group (Line 444) was not convincing. Just because leader support and family support prevented burnout among those working from home, doesn’t mean that support matters more. It may be that leader support and family support was not adequate enough for those at workplaces to prevent their burnout. Although both groups had increased engagement from coworker support and family support. At best these findings are mixed.
Specific Comments
P3 Line 110 “Thus researchers… concerned with the influence of workplace change on employee well-being during COVID” – this is where the groups can be clarified regarding “Work from Home and Change” and “Work from Workplace and No Change”. It is apparent that the authors are trying to measure two phenomena in one group “working from home” AND “experiencing a change in work arrangements due to the pandemic”. Please clarify how this can be?
P4 Line 198 “Data from Thailand..” this sentence seems unclear and not immediately relevant to the research questions guiding this paper. Do “perceived uncertainties” mediate the negative effect of supervisor support on exhaustion or does supervisor support mediate the negative effects of perceived uncertainty on exhaustion?
Author Response
Please find our response to your comments in the attached word file.
Best Regards,
Silje Fladmark
